# FASTER NO-REGRET LEARNING DYNAMICS FOR EXTENSIVE-FORM CORRELATED EQUILIBRIUM

## ABSTRACT

A recent emerging trend in the literature on learning in games has been concerned with providing accelerated learning dynamics for correlated and coarse correlated equilibria in normal-form games. Much less is known about the significantly more challenging setting of extensive-form games, which can capture sequential and simultaneous moves, as well as imperfect information. In this paper, we develop faster no-regret learning dynamics for *extensive-form correlated equilibrium (EFCE)* in multiplayer general-sum imperfect-information extensive-form games. When all agents play $T$ repetitions of the game according to the accelerated dynamics, the correlated distribution of play is an $O(T^{-3/4})$-approximate EFCE. This significantly improves over the best prior rate of $O(T^{-1/2})$. One of our conceptual contributions is to connect predictive (that is, optimistic) regret minimization with the framework of $\Phi$-regret. One of our main technical contributions is to characterize the stability of certain fixed point strategies through a refined perturbation analysis of a structured Markov chain, which may be of independent interest. Finally, experiments on standard benchmarks corroborate our findings.

## 1 INTRODUCTION

Game-theoretic solution concepts describe how agents should rationally act in games. Over the last two decades there has been tremendous progress in imperfect-information game solving and algorithms based on game-theoretic solution concepts have become the state of the art. Prominent milestones of this were an optimal strategy for Rhode Island hold'em poker (Gilpin & Sandholm, 2007), a near-optimal strategy for limit Texas hold'em (Bowling et al., 2015), and a superhuman strategy for no-limit Texas hold'em (Brown & Sandholm, 2017). In particular, these advances rely on algorithms that approximate *Nash equilibria* (*NE*) of two-player zero-sum *extensive-form games* (*EFGs*). EFGs are a broad class of games that capture sequential and simultaneous interaction, and imperfect information. For two-player zero-sum EFGs, it is by now well-understood how to compute a Nash equilibrium at scale: in theory this can be achieved using accelerated uncoupled no-regret learning dynamics, for example by having each player use an *optimistic* regret minimizer and leveraging suitable *distance-generating functions* (Hoda et al., 2010; Kroer et al., 2020; Farina et al., 2021c) for the EFG decision space. Such a setup converges to an equilibrium at a rate of $O(T^{-1})$. In practice, modern variants of the *counterfactual regret minimization* (*CFR*) framework typically lead to better performance, although the worst-case convergence rate is $O(T^{-1/2})$ (Zinkevich et al., 2007). CFR is also an uncoupled no-regret learning dynamic.

However, many real-world applications are not two-player zero-sum games, but instead have *general-sum* utilities and often more than two players. In such settings, Nash equilibrium suffers from several drawbacks when used as a prescriptive tool. First, there can be multiple equilibria, and an equilibrium strategy may perform very poorly when played against the "wrong" equilibrium strategies of the other player(s). Thus, the players effectively would need to communicate in order to find an equilibrium, or hope to converge to it via some sort of learning dynamics. Second, finding a Nash equilibrium is computationally hard both in theory (Daskalakis et al., 2006; Etessami & Yannakakis, 2007) and in practice (Berg & Sandholm, 2017). This effectively squashes any hope of developing efficient learning dynamics that converge to general-sum Nash equilibria.

A competing notion of rationality proposed by Aumann (1974) is that of *correlated equilibrium* (*CE*), typically modeled via a trusted mediator who privately recommends actions to the players.

Unlike NE, it is known that the latter can be computed in polynomial time and, perhaps even more importantly, it can be attained through *uncoupled* learning dynamics, where the players only need to reason about their own observed utilities. This overcomes the often unreasonable presumption that players have knowledge about the other players' utilities. At the same time, uncoupled learning algorithms have proven to be a remarkably *scalable* approach for computing equilibria in large-scale games, as described above. The basic CE notion is defined for normal-form games, and there it has long been known that uncoupled no-regret learning dynamics can converge to CE or the *coarse correlated equilibrium* (*CCE*) variant at a rate of $O(T^{-1/2})$ (Hart & Mas-Colell, 2000; Celli et al., 2019). More recently, it was shown that accelerated uncoupled no-regret learning dynamics can compute CCE and CE at a rate of $O(T^{-3/4})$ (Syrgkanis et al., 2015; Chen & Peng, 2020).

In the context of EFGs, the idea of correlation is much more intricate, and there are several notions of correlated equilibrium, based on when the mediator gives recommendations and how the mediator reacts to players who disregard the advice. One of the most compelling notions for EFGs is the *extensive-form correlated equilibrium* (henceforth EFCE) (von Stengel & Forges, 2008) for extensive-form games with *perfect recall*. Because of the sequential nature, the presence of private information in the game, and the gradual revelation of recommendations, the constraints associated with EFCE are significantly more complex than for normal-form games. For these reasons, the question of whether uncoupled learning dynamics can converge to an EFCE was only very recently resolved by Celli et al. (2020). Moreover, in a follow-up work they also established an explicit rate of convergence of $O(T^{-1/2})$ (Farina et al., 2021a). Our paper is concerned with the following fundamental question: *Can one develop faster uncoupled no-regret learning dynamics for* EFCE*?*

**Contributions.** Our primary contribution is to answer this question in the positive:

**Theorem 1.1.** *On any finite perfect-recall general-sum multiplayer extensive-form game, the uncoupled no-regret learning dynamics described in this paper lead to a correlated distribution of play that is an $O(T^{-3/4})$-approximate* EFCE*, where the $O(\cdot)$ notation suppresses game-specific parameters polynomial in the size of the game.*

We achieve this result using the framework of *predictive* (also known as *optimistic*) regret minimization (Chiang et al., 2012; Rakhlin & Sridharan, 2013b). One of our conceptual contributions is to connect this line of work with the framework of $\Phi$-*regret* minimization of Greenwald & Jafari (2003); Gordon et al. (2008), by providing a general template for stable-predictive $\Phi$-regret minimization. The importance of $\Phi$-regret is that it leads to substantially more powerful notions of hindsight rationality, beyond the usual *external* regret (Gordon et al., 2008), including the powerful notion of *swap regret* (Blum & Mansour, 2007). Moreover, one of the primary insights behind the result of Farina et al. (2021a) is to cast convergence to an EFCE as a $\Phi$-regret minimization problem. Given these prior connections, we believe that our stable-predictive $\Phi$ template is of independent interest, and could lead to further applications in the future.

Theorem 1.1 extends and strengthens several prior papers in the literature, including the seminal work of Syrgkanis et al. (2015) that provides accelerated dynamics for *coarse* correlated equilibrium in normal-form games, as well as the more recent result of Chen & Peng (2020) which showed $O(T^{-3/4})$ convergence to a correlated equilibrium in normal-form games. For the more challenging class of extensive-form games, accelerated rates were previously known only for finding a *Nash* equilibrium in the special case of *two-player zero-sum* games, where an $O(T^{-3/4})$ rate was achieved via a stable-predictive CFR setup (Farina et al., 2019a) and an $O(T^{-1})$ rate was achieved via optimistic regret minimizers coupled with good distance-generating functions (Farina et al., 2019c).

From a technical standpoint, in order to apply our generic template for accelerated $\Phi$-regret minimization, we establish two separate ingredients. First, we develop a *stable-predictive* external regret minimizer for the set of transformations $\Phi$ associated with EFCE. This differs from the construction by Farina et al. (2021a) in that we have to additionally guarantee and preserve the stability—and subsequently the predictivity—throughout the construction. The second component consists of sharply characterizing the stability of fixed points of *trigger deviation functions*. This turns out to be particularly challenging, and direct extensions of prior techniques appear to only give a bound that is *exponential* in the size of the game. In this context, one of our key technical contributions is to provide a refined perturbation analysis for a Markov chain consisting of a rank-one stochastic matrix, employing tools that have not been used before in this line of work, and substantially extending the techniques of Chen & Peng (2020). This leads to a rate of convergence that depends *polynomially*

on the description of the game, which is crucial for the applicability of the accelerated dynamics. Finally, we support our theoretical findings with experiments on several general-sum benchmarks.

**Further Related Work.** The line of work on accelerated no-regret learning for *Nash* equilibrium was pioneered by Daskalakis et al. (2015), showing that one can bypass the adversarial $\Omega(T^{-1/2})$ barrier for the incurred average regret if *both* players in a zero-sum game employ an uncoupled variant of the excessive gap technique (Nesterov, 2005), leading to a near-optimal rate of $O(\log T/T)$. Subsequently, Rakhlin & Sridharan (2013a) showed that the optimal rate of $O(1/T)$ can be obtained with a remarkably simple variant of Online Mirror Descent which incorporates a *prediction* term in the update step. While these results only hold for zero-sum games, Syrgkanis et al. (2015) showed that $O(T^{-3/4})$ rate can be obtained for multiplayer general-sum normal-form games. In a recent result, Chen & Peng (2020) strengthened the regret bounds of Syrgkanis et al. (2015) from external to swap regret using the celebrated construction of Blum & Mansour (2007). We also acknowledge a recent result of Daskalakis et al. (2021) which establishes a near-optimal rate of convergence of $\widetilde{O}(1/T)$ to a coarse correlated equilibrium when all players employ the Optimistic Multiplicative Weights Update (OMWU) algorithm in a normal-form game. Extending their result to extensive-form games presents considerable technical challenges since their analysis crucially hinges on the closed-form softmax-type structure of OMWU on the simplex.

Correlated equilibrium in extensive-form games is much less understood than Nash equilibrium. A feasible EFCE can also be computed efficiently through a variant of the *Ellipsoid algorithm* (Papadimitriou & Roughgarden, 2008; Jiang & Leyton-Brown, 2015), and an alternative sampling-based approach was given by Dudík & Gordon (2009). However, those approaches perform poorly in large-scale problems, and do not allow the players to arrive at EFCE via distributed learning. Celli et al. (2019) devised variants of the CFR algorithm that provably convergence to *normal-form coarse correlated equilibria*, a solution concept much less appealing than EFCE in extensive-form games Gordon et al. (2008). Finally, Morrill et al. (2021a;b) characterize hindsight rationality notions and associate a set of solution concepts with suitable $O(T^{-1/2})$ no-regret learning dynamics.

## 2 PRELIMINARIES

**Extensive-form Games.** An extensive-form game is abstracted on a directed and rooted *game tree* $\mathcal{T}$. The set of nodes of $\mathcal{T}$ is denoted with $\mathcal{H}$; non-terminal nodes are referred as *decision nodes*, and are associated with a player who acts by selecting an action from a set of possible actions $\mathcal{A}(h)$, where $h \in \mathcal{H}$ represents the decision node. By convention, the set of players $[n] \cup \{c\}$ includes a *fictitious* agent $c$ who "selects" actions according to fixed probability distributions dictated by the nature of the game (e.g., the roll of a dice); this intends to model external stochastic phenomena occurring during the game. For a player $i \in [n] \cup \{c\}$, we let $\mathcal{H}^{(i)} \subseteq \mathcal{H}$ be the subset of decision nodes wherein a player $i$ makes a decision. The set of *leaves* $\mathcal{Z} \subseteq \mathcal{H}$, or equivalently the *terminal nodes*, correspond to different outcomes; once the game transitions to a terminal node $z \in \mathcal{Z}$, payoffs are assigned to each player based on a set of *normalized* utility functions $\{u^{(i)} : \mathcal{Z} \to [-1, 1]\}_{i \in [n]}$. It will also be convenient to represent with $p^{(c)}(z)$ the product of probabilities of "chance" moves encountered in the path from the root until the terminal node $z \in \mathcal{Z}$.

*Imperfect Information.* To model imperfect information, the set of decision nodes $\mathcal{H}^{(i)}$ of player $i$ are partitioned into a collection of sets $\mathcal{J}^{(i)}$, which are called *information sets*. Each information set $j \in \mathcal{J}^{(i)}$ groups nodes which cannot be distinguished by $i$. Thus, for any nodes $h, h' \in j$ we have $\mathcal{A}(h) = \mathcal{A}(h')$. As usual, we assume that the game satisfies *perfect recall*: players never forget information once acquired. We will also define a partial order $\prec$ on $\mathcal{J}^{(i)}$, so that $j \prec j'$, for $j, j' \in \mathcal{J}^{(i)}$, if there exist nodes $h \in j$ and $h' \in j'$ such that the path from the root to $h'$ passes through $h$. If $j \prec j'$, we will say that $j$ is an *ancestor* of $j'$, or equivalently, $j$ is a descendant of $j'$.

*Sequence-form Strategies.* For a player $i \in [n]$, an information set $j \in \mathcal{J}^{(i)}$, and an action $a \in \mathcal{A}(j)$, we will denote with $\sigma = (j, a)$ the *sequence* of $i$'s actions encountered on the path from the root of the game until (and included) action $a$. For notational convenience, we will use the special symbol $\varnothing$ to denote the *empty sequence*. Then, $i$'s set of sequences is defined as $\Sigma^{(i)} := \{(j, a) : j \in \mathcal{J}^{(i)}, a \in \mathcal{A}(j)\} \cup \{\varnothing\}$; we will also use the notation $\Sigma_*^{(i)} := \Sigma^{(i)} \setminus \{\varnothing\}$. For a given information set $j \in \mathcal{J}^{(i)}$ we will use $\sigma^{(i)}(j) \in \Sigma^{(i)}$ to represent the *parent sequence*; i.e. the last sequence

encountered by player $i$ before reaching any node in the information set $j$, assuming that it exists. Otherwise, we let $\sigma^{(i)}(j) = \varnothing$, and we say that $j$ is the *root information set* of player $i$. A *strategy* for a player specifies a probability distribution for every possible information set encountered in the game tree. For perfect-recall EFGs, strategies can be equivalently represented in *sequence-form*:

**Definition 2.1** (Sequence-form Polytope). The *sequence-form strategy polytope* for player $i \in [n]$ is defined as the following (convex) polytope:

$$\mathcal{Q}^{(i)} := \left\{ \boldsymbol{q} \in \mathbb{R}_{\geq 0}^{|\Sigma^{(i)}|} : \boldsymbol{q}[\varnothing] = 1, \quad \boldsymbol{q}[\sigma^{(i)}(j)] = \sum_{a \in \mathcal{A}(j)} \boldsymbol{q}[(j,a)], \quad \forall j \in \mathcal{J}^{(i)} \right\}. \quad (1)$$

Analogously, one can define the sequence-form strategy polytope for the *subtree* of the partially ordered set $(\mathcal{J}^{(i)}, \prec)$ *rooted* at $j \in \mathcal{J}^{((i)}$, which will be denoted as $\mathcal{Q}_j^{(i)}$. Moreover, the set of *deterministic* sequence-form strategies for player $i \in [n]$ is the set $\Pi^{(i)} = \mathcal{Q}^{(i)} \cap \{0,1\}^{|\Sigma^{(i)}|}$, and similarly for $\Pi_j^{(i)}$. The *joint* set of deterministic sequence-form strategies of the players will be represented with $\Pi := \bigtimes_{i \in [n]} \Pi^{(i)}$. As such, an element $\boldsymbol{\pi} \in \Pi$ is an $n$-tuple $(\boldsymbol{\pi}^{(1)}, \ldots, \boldsymbol{\pi}^{(n)})$ specifying a deterministic sequence-form strategy for every player $i \in [n]$. Finally, the utility of player $i \in [n]$ under a profile $\boldsymbol{\pi} \in \Pi$ can be expressed as

$$u^{(i)}(\boldsymbol{\pi}) := \sum_{z \in \mathcal{Z}} p^{(c)}(z) u^{(i)}(z) \mathbf{1}\{\boldsymbol{\pi}^{(k)}[\sigma^{(k)}(z)] = 1, \forall k \in [n]\}. \quad (2)$$

We summarized in Table 1 the EFG notation that we will be using most often throughout the paper.

*An Illustrative Example.*   To clarify some of the concepts we have introduced, we illustrate a simple two-player EFG in Figure 1. Black nodes belong to player 1, white round nodes to player 2, square nodes are terminal nodes (aka leaves), and the crossed node is a chance node. Player 2 has two information sets, $\mathcal{J}^{(2)} := \{C, D\}$, each containing two nodes. This captures the lack of knowledge regarding the action played by player 1. In contrast, the outcome of the chance move is observed by both players. At the information set C, player 2 has two possible actions, $\mathcal{A}(C) := \{5, 6\}$. Thus, one possible sequence for player 2 is the pair $\sigma = (C, 5) \in \Sigma^{(2)}$.

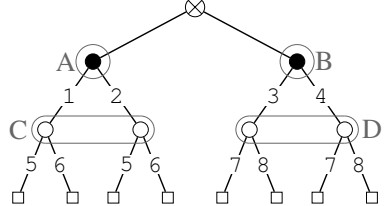

Figure 1: Example of a two-player EFG.

| | Description |
|---|---|
| $\mathcal{J}^{(i)}$ | Information sets of player $i$ |
| $\mathcal{A}(j)$ | Actions at information set $j$ |
| $\Sigma^{(i)}$ | Set of sequences of player $i$ |
| $\mathcal{Q}_j^{(i)}$ | Sequence-form strategies rooted at $j \in \mathcal{J}^{(i)}$ |
| $\mathfrak{D}^{(i)}$ | Maximum depth of any $j \in \mathcal{J}^{(i)}$ |

Table 1: Summary of the basic notation.

**Regret, $\Phi$-Regret and Optimistic Regret Minimization.**   Consider a convex and compact set $\mathcal{X} \subseteq \mathbb{R}^d$ representing the space of strategies of some agent. In the online decision making framework, a *regret minimizer* $\mathcal{R}$ can be thought of as a black-box device which interacts with the external environment via the following two basic subroutines:

- $\mathcal{R}$. NEXTSTRATEGY(): The regret minimizer returns the strategy $\boldsymbol{x}^t \in \mathcal{X}$ at time $t$;
- $\mathcal{R}$. OBSERVEUTILITY($\ell^t$): The regret minimizer receives as feedback a linear utility function $\ell^t : \mathcal{X} \ni \boldsymbol{x} \mapsto \langle \ell^t, \boldsymbol{x} \rangle$, and may alter its internal state accordingly.

The decision making is *online* in the sense that the regret minimizer can adapt to previously received information, but no information about future utilities is available. The error of a regret minimizer is typically measured in terms of *external regret*, defined, for a time horizon $T$, as follows:

$$R^T := \max_{\boldsymbol{x}^* \in \mathcal{X}} \sum_{t=1}^T \langle \boldsymbol{x}^*, \ell^t \rangle - \sum_{t=1}^T \langle \boldsymbol{x}^t, \ell^t \rangle, \quad (3)$$

That is, the performance of the online algorithm is compared with the best *fixed* strategy in *hindsight*.

$\Phi$-*Regret.* A conceptual generalization of the concept of external regret is the so-called $\Phi$-*regret*. Specifically, in this framework the performance of the learning algorithm is measured based on a *set of transformations* $\Phi : \mathcal{X} \to \mathcal{X}$, leading to the notion of cumulative $\Phi$-regret:

$$R^T := \max_{\phi^* \in \Phi} \sum_{t=1}^{T} \langle \phi^*(\boldsymbol{x}^t), \boldsymbol{\ell}^t \rangle - \sum_{t=1}^{T} \langle \boldsymbol{x}^t, \boldsymbol{\ell}^t \rangle. \tag{4}$$

When the set of transformations $\Phi$ coincides with the set of *constant* functions, one recovers the notion of external regret given in Equation (3). However, $\Phi$-regret is substantially more expressive and yields a more appealing notion of hindsight rationality (Gordon et al., 2008), incorporating the notion of *swap regret* (Blum & Mansour, 2007).

We will employ the following definition, which is a slight modification of the `RVU` property introduced by (Syrgkanis et al., 2015, Definition 3).

**Definition 2.2** (Stable-predictivity)**.** Let $\mathcal{R}$ be a regert minimizer and let $\| \cdot \|$ be a norm. $\mathcal{R}$ is said to be $\kappa$-*stable* with respect to $\| \cdot \|$ if for all $t \geq 2$, the strategies output by $\mathcal{R}$ satisfy

$$\|\boldsymbol{x}^t - \boldsymbol{x}^{t-1}\| \leq \kappa, \tag{5}$$

Moreover, it is said to be $(\alpha, \beta)$-*predictive* with respect to $\| \cdot \|$ if for all $t \geq 1$ its regret $R^T$ satisfies

$$R^T \leq \alpha(T) + \beta \sum_{t=2}^{T} \|\boldsymbol{\ell}^t - \boldsymbol{\ell}^{t-1}\|_*^2, \tag{6}$$

no matter the sequence of utility vectors $\boldsymbol{\ell}^1, \dots, \boldsymbol{\ell}^T$, where $\| \cdot \|_*$ is the dual norm of $\| \cdot \|$.

*Optimistic Follow the Regularized Leader.* Let $d$ be a 1-strongly convex function with respect to some norm $\| \cdot \|$, and $\eta > 0$ the *learning rate*. OFTRL's update rule takes the following form:

$$\boldsymbol{x}^t := \arg\max_{\boldsymbol{x} \in \mathcal{X}} \left\{ \left\langle \boldsymbol{x}, 2\boldsymbol{\ell}^{t-1} + \sum_{\tau=1}^{t-2} \boldsymbol{\ell}^\tau \right\rangle - \frac{d(\boldsymbol{x})}{\eta} \right\}, \tag{OFTRL}$$

where $\boldsymbol{x}^1 := \arg\min_{\boldsymbol{x} \in \mathcal{X}} d(\boldsymbol{x})$. Syrgkanis et al. (2015) established the following property:

**Lemma 2.3.** *(OFTRL) is $2\eta$-stable and $(\Omega_d/\eta, \eta)$-predictive with respect to any norm $\| \cdot \|$ for which $d$ is 1-strongly convex, where $\Omega_d$ is the range of $d$ on $\mathcal{X}$, that is, $\Omega_d := \max_{\boldsymbol{x}, \boldsymbol{x}' \in \mathcal{X}} \{d(\boldsymbol{x}) - d(\boldsymbol{x}')\}$.*

In this paper, we consider the entropic regularizer with respect to the simplex $d(\boldsymbol{x}) := \sum_{i=1}^{d} \boldsymbol{x}_i \log \boldsymbol{x}_i$, which is 1-strongly convex with respect to the $\ell_1$ norm. The pair of dual norms in the predictivity bound will therefore be $(\| \cdot \|_1, \| \cdot \|_\infty)$. We call this OFTRL setup *Optimistic Multiplicative Weights Updates* (OMWU).

**Extensive-Form Correlated Equilibrium.** We will work with the definition of EFCE due to Farina et al. (2019e), which is equivalent to that of von Stengel & Forges (2008). First, let us introduce the concept of a *trigger deviation function*.

**Definition 2.4.** Consider some player $i \in [n]$, a sequence $\hat{\sigma} = (j, a) \in \Sigma_*^{(i)}$, and joint sequence-form strategies $\boldsymbol{\pi} \in \Pi_j^{(i)}$. A *trigger deviation function* with respect to a *trigger sequence* $\hat{\sigma}$ and *continuation strategy* $\hat{\boldsymbol{\pi}}$ is any linear function $f : \mathbb{R}^{|\Sigma^{(i)}|} \to \mathbb{R}^{|\Sigma^{(i)}|}$ with the following properties.

- Any strategy $\boldsymbol{\pi} \in \Pi^{(i)}$ which does not prescribe the sequence $\hat{\sigma}$ remains invariant. That is, $f(\boldsymbol{\pi}) = \boldsymbol{\pi}$ for any $\boldsymbol{\pi} \in \Pi^{(i)}$ such that $\boldsymbol{\pi}[\hat{\sigma}] = 0$;
- Otherwise, the prescribed sequence $\hat{\sigma} = (j, a)$ is modified so that the behavior at $j$, as well as all its descendants is replaced by the behavior specified by the continuation strategy:

$$f(\boldsymbol{\pi})[\sigma] = \begin{cases} \boldsymbol{\pi}[\sigma] & \text{if } \sigma \not\succeq j; \\ \hat{\boldsymbol{\pi}}[\sigma] & \text{if } \sigma \succeq j, \end{cases} \tag{7}$$

for all $\sigma \in \Sigma^{(i)}$ and $\boldsymbol{\pi} \in \Pi^{(i)}$ such that $\boldsymbol{\pi}[\hat{\sigma}] = 1$.

We will let $\Psi^{(i)} := \{\phi^{(i)}_{\hat{\sigma} \to \hat{\pi}} : \hat{\sigma} = (j, a) \in \Sigma^{(i)}_*, \hat{\pi} \in \Pi^{(i)}_j\}$ be the set of all possible linear mappings defining trigger deviation functions for player $i$. We are ready to introduce the concept of EFCE.

**Definition 2.5** (EFCE). For $\epsilon \geq 0$, a probability distribution $\mu \in \Delta^{|\Pi|}$ is an $\epsilon$-approximate EFCE if for every player $i \in [n]$ and every trigger deviation function $\phi^{(i)}_{\hat{\sigma} \to \hat{\pi}} \in \Psi^{(i)}$, it holds that

$$\mathbb{E}_{\pi \sim \mu}\left[u^{(i)}\left(\phi^{(i)}_{\hat{\sigma} \to \hat{\pi}}(\pi^{(i)}), \pi^{(-i)}\right) - u^{(i)}(\pi)\right] \leq \epsilon, \tag{8}$$

where $\pi = (\pi_1, \ldots, \pi_n) \in \Pi$. A probability distribution $\mu \in \Delta^{|\Pi|}$ is an EFCE if it is a 0-EFCE.

**Theorem 2.6** (Farina et al. (2021a)). *For every player $i \in [n]$, let $\pi^{(i),1}, \ldots, \pi^{(i),T} \in \Pi^{(i)}$ be a sequence of deterministic sequence-form strategies whose cumulative $\Psi^{(i)}$-regret is $R^{(i),T}$ with respect to the sequence of linear utility functions*

$$\ell^{(i),t} : \Pi^{(i)} \ni \pi^{(i)} \mapsto u^{(i)}\left(\pi^{(i)}, \pi^{(-i),t}\right). \tag{9}$$

*Then, the empirical frequency of play $\mu \in \Delta^{|\Pi|}$ is an $\epsilon$-EFCE, where $\epsilon := \frac{1}{T} \max_{i \in [n]} R^{(i),T}$.*

## 3 ACCELERATING $\Phi$-REGRET MINIMIZATION VIA OPTIMISM

In this section we develop a general template for accelerated $\Phi$-regret minimization for general sets, and then we instantiate the template for dynamics for EFCE. Our approach combines a framework of Gordon et al. (2008) with the framework of stable-predictive (aka. optimistic) regret minimization. As in Gordon et al. (2008), in our template we combine 1) a regret minimizer that outputs a linear transformation $\phi^t \in \Phi$ at every time $t$, and 2) a fixed-point oracle for each $\phi^t \in \Phi$. However, in our framework, we further require that 2) is stable (in the sense of Definition 2.2). To achieve this, we will focus on regret minimizers that have the following property:

**Definition 3.1.** Consider a set of functions $\Phi$ such that $\phi(\mathcal{X}) \subseteq \mathcal{X}$ for all $\phi \in \Phi$, and a no-regret algorithm $\mathcal{R}_\Phi$ for the set of transformations $\Phi$ which returns a sequence $\phi^t \in \Phi$. We say that $\mathcal{R}_\Phi$ is *fixed point $G$-stable*, for $G \geq 0$, if the following conditions hold:

- Every $\phi^t$ admits a fixed point. That is, there exists $x^t \in \mathcal{X}$ such that $\phi^t(x^t) = x^t$.
- For any $x^t$ such that $x^t = \phi^t(x^t)$, there exists $x^{t+1}$ with $x^{t+1} = \phi^{t+1}(x^{t+1})$ such that $\|x^{t+1} - x^t\| \leq G$.

We will show how to construct an accelerated $\Phi$-regret minimizer starting from the following:

1. $\mathcal{R}_\Phi$: A $\kappa$-stable $(\alpha, \beta)$-predictive fixed point $G$-stable regret minimizer for $\Phi$;
2. STABLEFPORACLE($\phi; \widetilde{x}, G, \epsilon$): A *stable fixed point oracle* which returns a point $x \in \mathcal{X}$ such that (i) $\|\phi(x) - x\| \leq \epsilon$, and (ii) $\|x - \widetilde{x}\| \leq G$ (the existence of such a fixed point is guaranteed by the fixed point $G$-stability assumption for the regret minimizer).

Given these two components, our next theorem builds a stable-predictive $\Phi$-regret minimizer.

**Theorem 3.2** (Accelerated $\Phi$-Regret Minimization). *Consider a $\kappa$-stable $(\alpha, \beta)$-predictive regret minimizer $\mathcal{R}_\Phi$ for a set of linear transformations $\Phi$, with respect to the $\ell_1$ norm $\|\cdot\|_1$. Moreover, assume that $\mathcal{R}_\Phi$ is fixed point $G$-stable with respect to $\Phi$. Then, if we have access to a STABLEFPORACLE, we can construct a $G$-stable algorithm with $\Phi$-regret $R^T$ bounded as*

$$R^T \leq \alpha(T) + 2\beta D_\ell^2 \kappa^2 T + 2\beta \sum_{t=2}^T \|\ell^t - \ell^{t-1}\|_\infty^2 + D_\ell \sum_{t=1}^T \epsilon_t, \tag{10}$$

*where $\epsilon_t$ is the error of STABLEFPORACLE at time $t$, and $D_\ell$ is an upper bound on the $\ell_\infty$ norm of $\ell^t$'s. It is also assumed that $\|x\|_\infty \leq 1$ for all $x \in \mathcal{X}$.*

The proof is similar to that of Gordon et al. (2008), and is included in Appendix B.

### 3.1 CONSTRUCTING A STABLE-PREDICTIVE REGRET MINIMIZER FOR $\Psi^{(i)}$

Here we develop a regret minimizer for the set $\text{co } \Psi^{(i)}$, the convex hull of the set of trigger deviation functions. Given that $\text{co } \Psi^{(i)} \supseteq \Psi^{(i)}$, this will immediately imply a regret minimizer for the set

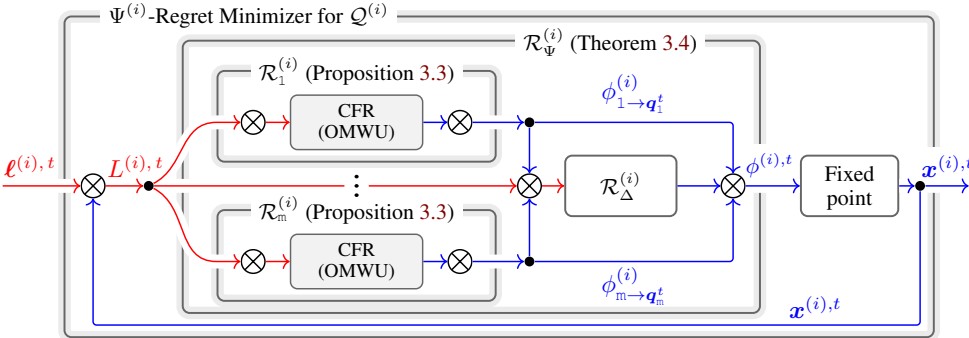

Figure 2: An overview of the overall construction. For notational convenience we have let $\Sigma_*^{(i)} := \{1, 2, \ldots, \mathtt{m}\}$. The symbol $\otimes$ in the figure denotes a multilinear transformation of the inputs. We also note that blue corresponds to the iterates, while red corresponds to the utilities.

$\Psi^{(i)}$. An overview of the algorithm is given in Figure 2. Farina et al. (2021a) observed that the set $\operatorname{co} \Psi^{(i)}$ can be evaluated in two stages. First, for a fixed sequence $\hat{\sigma} = (j, a) \in \Sigma_*^{(i)}$ we define the set $\Psi_{\hat{\sigma}}^{(i)} := \operatorname{co}\left\{\phi_{\hat{\sigma} \to \hat{\pi}} : \hat{\pi} \in \Pi_j^{(i)}\right\}$; then, we take the convex hull of all $\Psi_{\hat{\sigma}}^{(i)}$, that is, $\operatorname{co} \Psi^{(i)} = \operatorname{co}\{\Psi_{\hat{\sigma}}^{(i)} : \hat{\sigma} \in \Sigma_*^{(i)}\}$. Correspondingly, we first develop a stable-predictive regret minimizer for the set $\Psi_{\hat{\sigma}}^{(i)}$, for any $\hat{\sigma} \in \Sigma_*^{(i)}$, and these individual regret minimizers are then combined using a *regret circuit* to conclude the construction in Theorem 3.4. All the omitted poofs and pseudocode for this section are included in Appendix B.1.

**Stable-Predictive Regret Minimizer for the set $\Psi_{\hat{\sigma}}^{(i)}$.** Consider a sequence $\hat{\sigma} \in \Sigma_*^{(i)}$. Farina et al. (2021a) observed that the set of transformations $\Psi_{\hat{\sigma}}^{(i)} := \operatorname{co}\left\{\phi_{\hat{\sigma} \to \hat{\pi}} : \hat{\pi} \in \Pi_j^{(i)}\right\}$ is the image of $\mathcal{Q}_j^{(i)}$ under the affine mapping $h_{\hat{\sigma}}^{(i)} : \boldsymbol{q} \mapsto \phi_{\hat{\sigma} \to \boldsymbol{q}}^{(i)}$. Hence, it is well-known that a regret minimizer for $\Psi_{\hat{\sigma}}^{(i)}$ can be constructed starting from a regret minimizer for $\mathcal{Q}_j^{(i)}$. We now show that the same can be said if one restricts to *stable-predictive* regret minimizers. In particular, we have the following.

**Proposition 3.3.** *Consider a player $i \in [n]$ and any trigger sequence $\hat{\sigma} = (j, a) \in \Sigma_*^{(i)}$. There exists an algorithm which constructs a deterministic regret minimizer $\mathcal{R}_{\hat{\sigma}}^{(i)}$ with access to a $K$-stable $(A_T, B)$-predictive deterministic regret minimizer $\mathcal{R}_{\mathcal{Q}}^{(i)}$ for the set $\mathcal{Q}_j^{(i)}$, such that $\mathcal{R}_{\hat{\sigma}}^{(i)}$ is $K$-stable and $(A_T, B)$-predictive.*

In Appendix A we describe a stable-predictive variant of CFR for the set $\mathcal{Q}_j^{(i)}$, for each $j \in \mathcal{J}^{(i)}$, following the construction of Farina et al. (2019a).

**Stable-Predictive Regret Minimizer for $\operatorname{co} \Psi^{(i)}$.** The next step consists of combining the regret minimizers $\Psi_{\hat{\sigma}}^{(i)}$, for all $\hat{\sigma} \in \Sigma_*^{(i)}$, to a composite regret minimizer for the set $\operatorname{co} \Psi^{(i)}$. To this end, we employ *regret circuits* (Farina et al., 2019d), leading to the main result of this section:

**Theorem 3.4.** *Consider a $\kappa$-stable $(\alpha, \beta)$-predictive regret minimizer $\mathcal{R}_{\Delta}^{(i)}$ for the the simplex $\Delta^{|\Sigma_*^{(i)}|}$, and $K$-stable $(A, B)$-predictive regret minimizers $\mathcal{R}_{\hat{\sigma}}^{(i)}$ for each $\hat{\sigma} \in \Sigma_*^{(i)}$, all with respect to the pair of norms $(\| \cdot \|_1, \| \cdot \|_\infty)$. Then, there exists an algorithm which constructs a regret minimizer $\mathcal{R}_{\Psi}^{(i)}$ for the set $\operatorname{co} \Psi^{(i)}$ such that (i) $\mathcal{R}_{\Psi}^{(i)}$ is $O(K + |\Sigma^{(i)}|\kappa)$-stable, and (ii) under any sequence of linear utility functions $L^1, \ldots, L^T$ the regret incurred can be bounded as*

$$R_{\Psi}^T \leq O(\alpha(T) + A(T) + \beta D_{\mathbf{L}}^2 K^2 T) + O(B + \beta|\Sigma^{(i)}|^2) \sum_{t=2}^{T} \|\mathbf{L}^t - \mathbf{L}^{t-1}\|_\infty^2, \qquad (11)$$

*where $\|\mathbf{L}^t\|_\infty \leq D_{\mathbf{L}}$.*

## 3.2 STABILITY OF THE FIXED POINTS

In this subsection we complete the construction of the $\Psi^{(i)}$-regret minimizer by establishing a *stable* fixed point oracle for any $\phi \in \mathrm{co}\,\Psi^{(i)}$. All of the proofs of this section are included in Appendix B.2.

**Multiplicative Stability.** A sequence $\{z^t\}$, with $z^t \in \mathbb{R}_{\geq 0}^d$, is said to be $\kappa$-*multiplicative-stable* if $(1-\kappa)z_i^{t-1} \leq z_i^t \leq (1+\kappa)z_i^{t-1}$, for any $i \in [d]$, and for all $t \geq 2$. Importantly, this notion of multiplicative stability is guaranteed by OMWU (see Lemma B.2). Thus, if $\mathfrak{D}^{(i)}$ is the depth of $i$'s actions and $D_{\boldsymbol{x}}^{(i)}$ is an upper bound on the $\ell_1$ norm in the treeplex, we can show the following:

**Lemma 3.5.** *When each regret minimizer $\mathcal{R}_{\hat{\sigma}}^{(i)}$ is constructed using predictive* CFR *instantiated with* OMWU *with learning rate $\eta$ (Theorem A.4) such that for all $\hat{\sigma} \in \Sigma_*^{(i)}$, the output sequence is $O(\eta(\mathfrak{D}^{(i)})^2 D_{\boldsymbol{x}}^{(i)} D_{\boldsymbol{\ell}})$-multiplicatively-stable. Moreover, if the regret minimizer $\mathcal{R}_{\Delta}^{(i)}$ is realized using* OMWU *with learning rate $\eta$, it will output an $O(\eta|\Sigma^{(i)}|D_{\boldsymbol{\ell}})$-multiplicatively-stable sequence.*

This characterization will be crucial for establishing the stability of the fixed points. In particular, following the approach of Farina et al. (2021a), let us introduce the following definitions:

**Definition 3.6.** Consider a player $i \in [n]$ and let $J \subseteq \mathcal{J}^{(i)}$ be a subset of $i$'s information sets. We say than $J$ is a *trunk* of $\mathcal{J}^{(i)}$ if, for every $j \in J$, all predecessors of $j$ are also in $J$.

**Definition 3.7.** Consider a player $i \in [n]$, a trunk $J \subseteq \mathcal{J}^{(i)}$, and $\phi \in \mathrm{co}\,\Psi^{(i)}$. A vector $\boldsymbol{x} \in \mathbb{R}_{\geq 0}^{|\Sigma^{(i)}|}$ is a *$J$-partial fixed point* of $\phi$ if the following conditions hold:

- $\boldsymbol{x}[\varnothing] = 1$ and $\boldsymbol{x}[\sigma^{(i)}(j)] = \sum_{a \in \mathcal{A}(j)} \boldsymbol{x}[(j,a)]$, for all $j \in J$;
- $\phi(\boldsymbol{x})[\varnothing] = \boldsymbol{x}[\varnothing] = 1$, and $\phi(\boldsymbol{x})[(j,a)] = \boldsymbol{x}[(j,a)]$, for all $j \in J$, and $a \in \mathcal{A}(j)$.

An important property is that a $J$-partial fixed point can be efficiently "promoted" to a $J \cup \{j^*\}$-partial fixed point by computing the stationary distribution of a certain Markov chain. However, a significant concern is whether this fixed point operation can potentially cause a substantial degradation in terms of stability. One of our key results is that the associated Markov chain has a particular structure, which enables us to substantially improve the stability bound and thereby obtain a polynomial degradation in stability. More precisely, this boils down to the following technical lemma.

**Lemma 3.8.** *Let $\mathbf{M}$ and $\mathbf{M}'$ be transition matrices of $m$-state Markov chains such that $\mathbf{M} = \boldsymbol{v}\mathbf{1}^\top + \mathbf{C}$ and $\mathbf{M}' = \boldsymbol{v}'\mathbf{1}^\top + \mathbf{C}'$, where $\mathbf{C}, \mathbf{C}', \boldsymbol{v}, \boldsymbol{v}'$ have strictly positive entries. Moreover, let $\boldsymbol{\pi}$ and $\boldsymbol{\pi}'$ be the (unique) stationary distributions of $\mathbf{M}$ and $\mathbf{M}'$ respectively. Then, if (i) the entries of the matrices $\mathbf{C}$ and $\mathbf{C}'$ are $\kappa$-multiplicatively-close, (ii) the entries of the vectors $\boldsymbol{v}$ and $\boldsymbol{v}'$ are $\gamma$-multiplicatively-close, and (iii) the sum of the entries of $\boldsymbol{v}$ and $\boldsymbol{v}'$ are $\kappa$-multiplicatively-close, then $\boldsymbol{\pi}$ and $\boldsymbol{\pi}'$ are $(\gamma + O(\kappa m))$-multiplicatively-close, for a sufficiently small $\kappa = O(1/m)$.*

Using a slightly more general result (Corollary B.10), we manage to obtain the following:

**Proposition 3.9.** *Consider a player $i \in [n]$, and let $\phi = \sum_{\hat{\sigma} \in \Sigma_*^{(i)}} \boldsymbol{\lambda}[\hat{\sigma}] \phi_{\hat{\sigma} \to \boldsymbol{q}_{\hat{\sigma}}}^{(i)}$ be a transformation in $\mathrm{co}\,\Psi^{(i)}$ such that the sequence of $\boldsymbol{\lambda}^t$'s and $\boldsymbol{q}_{\hat{\sigma}}^t$'s is $\kappa$-multiplicatively-stable, for all $\hat{\sigma} \in \Sigma_*^{(i)}$. If $\boldsymbol{x}^t$ is a $\gamma$-multiplicatively-stable $J$-partial fixed point sequence, there is an algorithm which computes a $(J \cup \{j^*\})$-partial fixed point $(\boldsymbol{x}^t)'$ of $\phi$ such that the sequence of $(\boldsymbol{x}')^t$'s is $(\gamma + O(\kappa|\mathcal{A}(j^*)|))$-multiplicatively-stable, for any sufficiently small $\kappa = O(1/|\mathcal{A}(j^*)|)$.*

Thus, using our technical lemma, we manage to bypass the substantial overhead of the term $\gamma|\mathcal{A}(j^*)|$, which would follow using techniques similar to Chen & Peng (2020). This turns out to be crucial for obtaining a polynomial dependence on the size of the game. Finally, we can inductively employ this proposition to show the overall stability of the fixed points:

**Theorem 3.10.** *Consider a player $i \in [n]$, and let $\phi = \sum_{\hat{\sigma} \in \Sigma_*^{(i)}} \boldsymbol{\lambda}[\hat{\sigma}] \phi_{\hat{\sigma} \to \boldsymbol{q}_{\hat{\sigma}}}^{(i)}$ be a transformation in $\mathrm{co}\,\Psi^{(i)}$ such that the sequence of $\boldsymbol{\lambda}^t$'s and $\boldsymbol{q}_{\hat{\sigma}}^t$'s is $\kappa$-multiplicatively-stable, for all $\hat{\sigma} \in \Sigma_*^{(i)}$. Then, there exists an algorithm which computes a fixed point $\boldsymbol{q}^t \in \mathcal{Q}^{(i)}$ of $\phi$ such that the sequence of $\boldsymbol{q}^t$'s is $O(\kappa|\mathcal{A}^{(i)}|\mathfrak{D}^{(i)})$-multiplicatively-stable, where $|\mathcal{A}^{(i)}| := \max_{j \in \mathcal{J}^{(i)}} |\mathcal{A}(j)|$, and for a sufficiently small $\kappa = O(1/(|\mathcal{A}^{(i)}|\mathfrak{D}^{(i)}))$.*

Finally, if we use the stability values derived in Lemma 3.5, we arrive at the following conclusion:

**Corollary 3.11.** *For $\kappa = O((D_x^{(i)}(\mathfrak{D}^{(i)})^2 + |\Sigma^{(i)}|)|\mathcal{A}^{(i)}|\mathfrak{D}^{(i)}D_\ell)$, the sequence of fixed points will be $(\eta\kappa)$-multiplicatively-stable, for any sufficiently small $\eta = O(1/\kappa)$.*

*Putting Everything Together.*   Finally, having established these ingredients, we can use the template of Theorem 3.2 to obtain Theorem 1.1, as we formally show in Appendix B.3.

## 4   EXPERIMENTS

In this section we experimentally investigate the performance of our stable-predictive algorithm compared to two other popular approaches based on a CFR-style decomposition of regrets into local regret-minimization problems: the existing algorithm by Farina et al. (2021a) instantiated with (i) *regret matching*$^+$ (RM$^+$) (Tammelin, 2014) for each simplex (in place of regret matching), and (ii) using the vanilla MWU algorithm for each simplex. In accordance to the theoretical predictions, the stepsize for OMWU is set as $\eta_t = \tau \cdot t^{-1/4}$ (cf. Corollary B.13), and for MWU it is set as $\eta_t = \tau \cdot t^{-1/2}$, where the parameter $\tau$ is chosen by picking the best-performing value among $\{0.01, 0.1, 1, 10, 100\}$. In particular, we evaluate their performance based on the following popular benchmark games: (i) a three-player variant of *Kuhn poker* (Kuhn, 1950); (ii) a two-player bargaining game known as *Sheriff* (Farina et al., 2019e)—a benchmark game introduced specifically for the study of correlated equilibria; and (iii) a three-player version of *Liar's dice* (Lisý et al., 2015). A detailed description of each of the three game instances is available in Appendix D.

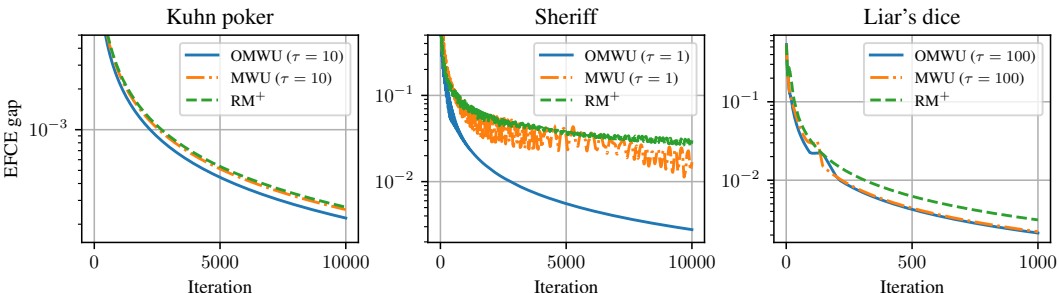

Figure 3: The performance of MWU, OMWU, and RM$^+$ on three general-sum EFGs.

Figure 3 shows the performance of each of the three learning dynamics for computing EFCE. On the $x$-axis we plot the number of iterations performed by each algorithm, and on the $y$-axis we plot the EFCE gap, defined as the maximum advantage that any player can gain by defecting optimally from the mediator's recommendations. It should be noted that one iteration costs the same for every algorithm, up to constant factors. We see that on every game, OMWU performs better than or on par with RM$^+$ and MWU. On Sheriff, OMWU performs significantly better than both RM$^+$ and MWU, by about an order of magnitude. One caveat to these results is that we did not use two tricks that help CFR$^+$ in two-player zero-sum EFG solving: alternation and linear averaging. These tricks are known to retain convergence guarantees in that context (Tammelin et al., 2015; Farina et al., 2019b; Burch et al., 2019), but it is unclear if they still guarantee convergence in the EFCE setting.

## 5   CONCLUSIONS

We described uncoupled no-regret learning dynamics so that if all agents play $T$ repetitions of the game according to the dynamics, the correlated distribution of play is an $O(T^{-3/4})$-approximate EFCE. This substantially improves over the prior best rate of $O(T^{-1/2})$. One of our conceptual contributions is to connect the line of work on optimistic regret minimization with the framework of $\Phi$-regret. One of our main technical contributions is to characterize the stability of the fixed points associated with trigger deviation functions through a refined perturbation analysis of a certain structured Markov chain, which may be of independent interest. Finally, experiments conducted on standard benchmarks corroborated our theoretical findings.

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
