# OpenReview forum: "Faster No-Regret Learning Dynamics for Extensive-Form Correlated Equilibrium"
_ICLR.cc/2022/Conference — ICLR 2022 Submitted_

### Official Review · Reviewer_mWDx · 2021-11-02

**Correctness:** 3
**Technical Novelty And Significance:** 2
**Empirical Novelty And Significance:** Not applicable
**Recommendation:** 5
**Confidence:** 4

**Main Review:**

On the positive side the improved convergence rate is a clear theoretical contribution and the experiments suggest that a practical implementation of these techniques is possible.  On the negative side, as discussed in detail below, the techniques used to achieve it essentially boil down to combining the approach of Farina et al. 2021a with a line of work on slowly-changing regret minimizers which have been shown to achieve accelerated rates in a number of settings, so the acceleration here is not terribly surprising.  While such a combination may require substantial novel insights, the exposition does not currently make it clear what those are.

More broadly, the positioning of the contribution of this paper relative to related work needs to be clearer.  I will discuss several particular relationships which do not appear to be sufficiently clearly explained, and for some of them this appears to expose weaknesses in the paper.

Farina et al. 2021b [F21b].  This paper achieves a O(T^{-0.5|) rate to a weaker solution concept (EFCCE).  However, relative to EFCE it avoids an expensive computation at each step.  This paper is used as a baseline in the experiments Section 4, and thus has two differences (slower rate and different solution concept).  The discussion omits the second difference and appears to discuss it as converging to an EFCE.  So I am left wondering which factor is responsible for the worse performance in the experiments.  Furthermore, Appendix C of the supplemental materials (which does not appear to be referenced in the main text) discusses how the result of this paper can be combined with those of [F21b] to get accelerated convergence to EFCCE.  Give this, I am not sure why the experiments do not isolate the two factors (rate vs solution concept) and which version of the algorithm in the paper the experiments are actually testing (i.e. do they actually use the EFCCE version?).

The line of work on distance-generating functions (Hoda et al. 2010, Kroer et al. 2020, Farina et al. 2021c).  These techniques have achieved O(T^{-1}) in other settings but are mentioned only briefly.  Why not adopt these techniques rather than the ones that lead to O(T^{-0.75})?  I don’t a-priori see why they should be incompatible with the framework from Section 3 and if they can be achieved it makes the headline result of O(T^{-0.75}) less impressive unless there are other reasons to prefer this approach.

Farina et al. 2021a [F21a].  While this work is referenced throughout the paper, the technical approaches share so much overlap that I have some difficulty disentangling where exactly the technical contributions in this paper are.  In particular, I am not precisely clear what constitutes the “stable predictive \Phi template” referred to as being of independent interest.  [F21a] appear to use the same framework based on Gordon et al. 2008 which incorporates an arbitrary regret minimizing dynamics.  So it would be helpful to call out which specific results require new insights to apply it to a stable-predictive version (which have been explored in several previous works, although perhaps mostly for normal-form games).  The one result specifically mentioned is Lemma 3.9, and I could not find a proof of it in the supplementary material (there does seem to be a proof of the reference Corollary B.9 but it builds on different results, although they appear at least somewhat similar in spirit).  As a result I have a hard time evaluating the significance of the technical contribution of this paper.  Finally, as a minor note I am a bit confused about the status of [F21a].  This paper describes it simply as “very-recent follow-up work” to Celli et al. 2020, while [F21a] describes itself as an extended version of that paper in a note on the first page.


**Summary Of The Paper:**

Previous work has identified uncoupled, regret-based dynamics that converge to extensive-form correlated equilibria in extensive-form games at a rate of O(T^{-0.5}) where T is the number of repetitions of the game.  This paper provides a new uncoupled, regret-based dynamics that achieves O(T^{-0.75}).  It does so by combining analysis techniques for regret-based dynamics that have shown such accelerated convergence in other settings with the framework used to achieve the  O(T^{-0.5}) rate.  Technical contributions in realizing this combination include a new analysis of a relevant Markov chain.  A limited set of experiments confirms faster convergence relative to prior approaches.

**Summary Of The Review:**

While I appreciate the headline theoretical contribution, given my concerns I am not convinced that this version of the paper makes a sufficient novel technical contribution.  One possibility is that it does but the technical exposition needs to be improved to more clearly highlight the contribution.  Another path forward, since this paper is heavily based on the approach of [F21a], improves its results, and that paper appears to be still unpublished, would be to consider whether the authors of {F21a] would be amenable to merging the papers.  A third possibility to extend the results and demonstrate the power and flexibility of the approach would be to show that it can also be adapted to work with the line of work which uses dynamics based on carefully-designed distance-generating functions to achieve an O(T^{-1}) bound.

---

> ### Author Response · Authors · 2021-11-13
> **Response to Reviewer mWDx**
>
> We are grateful to the reviewer for the comments and the suggestions. Below we give our response.
>
> --- "[...] This paper (Farina et al., 2021b) is used as a baseline in the experiments Section 4 [...]":
>
> There appears to be a misunderstanding caused by the fact that we accidentally cited Farina et al. (Efficient decentralized learning dynamicsfor extensive-form coarse correlated equilibrium: No expensive computation of stationary distributions required), instead of Farina et al. (Simple uncoupled no-regret learning dynamics for extensive-form correlated equilibrium) in the beginning of Section 4. We apologize to the reviewer for the confusion this has caused. To clarify this issue, let us stress that the primary focus of our paper lies on the solution concept of EFCE. For this reason, for our experiments we only measure convergence to EFCE; that is, convergence to the weaker notion of EFCCE is not investigated in our experiments. In particular, note that our algorithm is very different from the one used by Farina et al. (Efficient decentralized learning dynamicsfor extensive-form coarse correlated equilibrium: No expensive computation of stationary distributions required). As the reviewer points out, in Appendix C we also obtain slightly improved stability bounds for EFCCE, and without resorting to the involved techniques required for the stability of fixed points associated with EFCE, but Appendix C has been mostly included for the sake of completeness.
>
> --- "The line of work on distance-generating functions (Hoda et al. 2010, Kroer et al. 2020, Farina et al. 2021c). These techniques have achieved $O(T^{-1})$":
>
> The first-order methods based on distance-generating functions cited by the reviewer are not known to converge to EFCE for arbitrary multi-player general-sum EFGs. In particular, note that first-order methods only guarantee convergence to EFCE for restricted classes of games (e.g. two-player games with public chance moves). This is in contrast to our main result which applies to arbitrary EFGs.
>
> --- "In particular, I am not precisely clear what constitutes the “stable predictive $\Phi$ template” referred to as being of independent interest [...] I have a hard time evaluating the significance of the technical contribution of this paper":
>
> We have created a separate subsection in the introduction to better highlight our own contributions. In particular, in the third paragraph of the Contributions section (which has been highlighted) we have included additional discussion on our technical contributions and how our approach differs from prior results. We note that while the framework of $\Phi$-regret minimization is well-known, obtaining accelerated rates beyond the barrier of $T^{-1/2}$ requires further structural properties, namely in the form of the stability and subsequently the predictivity of the underlying regret minimizers. As a result, while the approach of Farina et al. illustrates how to cast the convergence to EFCE in the framework of $\Phi$-regret, there is still technical work to be done to ensure stability and predictivity.
>
> --- "The one result specifically mentioned is Lemma 3.9, and I could not find a proof of it in the supplementary material":
>
> Note that Lemma 3.9 is just an application of Corollary B.9 (Corollary B.10 in the revised version), and thus its proof follows the argument of Corollary B.9. For simplicity, we only state Lemma 3.9 in the main body since it captures our main technical refinement.
>
> --- "I am a bit confused about the status of [F21a]. This paper describes it simply as “very-recent follow-up work” to Celli et al. 2020, while [F21a] describes itself as an extended version of that paper in a note on the first page":
>
> The original work by Celli et al. 2020 did not provide a rate of convergence. In a subsequent work by the same authors, which indeed appears to be an extended version of the previous work, it was shown how to obtain a rate of $O(T^{-1/2})$.

---

> > ### Comment · Reviewer_mWDx · 2021-11-19
> > **Thank you for the response**
> >
> > The correction regarding the experiments addresses one of my significant concerns and does make me somewhat more positively disposed toward the paper.
> >
> > The new paragraph in the introduction does make it clearer what is new in the technical analysis relative to (Farina et al. 2021a)+(Chen and Peng 2020), but at least to a casual examination it isn't obvious to me that either of these really represent developing substantial new techniques.  The heavy lifting of guaranteeing stability and predictivity seems to be done by the framework of Farina et al. 2019d.  Chen and Peng also use the Markov chain tree theorem, and the main novelty seems to be using a variant of it due to Kruckman et al. 2010 to achieve the more refined bounds.  If this understanding is correct, and the main insight is to switch one version of the theorem for a variant, I don't know that I would describe it as "employing tools that have not been used before in this line of work" or a "substantial extension" of the techniques.
> >
> > This is not to say that executing on these changes was trivial, but if the parts that are truly new are minor, with most of the work being a careful synthesis of existing techniques, the contribution of this paper relative to Farina et al. 2021a seems a bit limited to me and I would reiterate my recommendation that one way forward with this paper would be to approach the authors of that paper about the possibility of merging them.

---

> > > ### Author Response · Authors · 2021-11-19
> > > **Response to Reviewer mWDx**
> > >
> > > Thanks for your comments.
> > >
> > > --- “the main novelty seems to be using a variant of it due to Kruckman et al. 2010 to achieve the more refined bounds”:
> > >
> > > This is not correct. Kruckman et al.’s technical report gives a proof of the classic Markov chain tree theorem—the same textbook theorem that was used by Chen & Peng. In particular, they do not prove any new variant. We simply found the linear-algebraic framework used by Kruckman et al. useful for our refined analysis of updated Markov chains, and more natural than other combinatorial approaches found elsewhere. When we said “Our approach is based on the techniques of Kruckman et al. (2010),” we were only referring to this part. We will improve the wording to clarify our contributions.
> > >
> > > Our refined analysis of the stability of stationary distributions of rank-one-updated Markov chains is not due to Kruckman et al., and our key lemma (Lemma B.8) gives a substantially different formula than the Markov chain tree theorem applied to the updated transition matrix M. In particular, in (36) the numerator depends linearly on the vector v, as opposed to the Markov chain tree theorem where terms of order v^n, where n is the number of states in the Markov chain, would appear both in the numerator and in the denominator, substantially undermining the analysis of the multiplicative stability of pi. We will make sure to clarify this point in the final version.
> > >
> > >
> > > --- “The heavy lifting of guaranteeing stability and predictivity seems to be done by the framework of Farina et al. 2019d”
> > >
> > > We’re not sure if there’s a misunderstanding here, but Farina et al. 2019d does not include any results on stability and predictivity. Rather, they give a general calculus for constructing regular (non-predictive and non-stable) regret minimizers for combinatorial sets. Perhaps the wording right before Theorem 3.4 was poorly chosen and did not accurately reflect the extensions needed in order to ensure and quantify predictivity and stability (we’ll improve that in the final version). Please see Equation (26) in the proof of Theorem 3.4; that is the part that comes from Farina et al 2019d.

---

> > > > ### Comment · Reviewer_mWDx · 2021-11-22
> > > > **Brief Followup**
> > > >
> > > > I understand that Farina et al. 2019d provides a general framework.  What still isn't clear, although perhaps the latest response hints at it, is to what extent novel techniques are required to get something that is stable and predictive in their framework.  If one sits down with this as a goal and a good understanding of the prior literature on stable and predictive approaches in other settings is this straightforward to achieve?  If not, why not?

---

> > > > > ### Author Response · Authors · 2021-11-23
> > > > > **Response to followup**
> > > > >
> > > > > To answer the reviewer's question, let us first stress that preserving and quantifying the stability-predictivity of our algorithm requires additional technical work and different techniques than the framework of Farina et al. (2019d)---which concerns regular (non-predictive) regret minimization, while it also differs from all prior works on stable-predictive regret minimization due to the challenging structure of the set of transformations comprising trigger deviation functions. Moreover, as we explained in our previous response, establishing multiplicative stability (especially under the fixed point computation associated with EFCE) proved to be a considerable challenge, and we had to develop new techniques (e.g., Lemma B.8).
> > > > >
> > > > > So we believe that proving our results is *not* "straightforward to achieve": our analysis required new technical contributions, on top of appropriately combining existing tools and techniques from the literature to establish our accelerated no-regret dynamics for EFCE---the first of their kind.

---

### Official Review · Reviewer_zmM5 · 2021-11-02

**Correctness:** 3
**Technical Novelty And Significance:** 3
**Empirical Novelty And Significance:** 3
**Recommendation:** 6
**Confidence:** 4

**Details Of Ethics Concerns:**

I do not have any concerns.

**Main Review:**

Computing game-theoretic solution concepts that capture the idea that the players may coordinate their strategies based on external signals has become an increasingly active field in recent years, whether in the form of correlated equilibrium, team correlated equilibrium, or other related concepts. Correlation has several attractive properties, with efficient computability among the most prominent ones and enough applications to justify further research in this direction. Designing uncoupled dynamics to approximate an extensive-form correlated equilibrium faster is hence definitely a problem worth studying. The authors' result is interesting, and I am convinced that the presented experimental results provide enough evidence to support their claims. I went through some of the proofs included in the appendix, and I could not find any obvious mistakes. The authors cite the relevant literature, and I also appreciate that they mention that alternation and linear averaging were left out from the experiments because of their lack of guarantees, which I find to be a valid argument. I have a few concerns as well, though.

Perhaps the main one relates to the presentation of this work. Explaining the problem and techniques the authors study requires a lot of background. Already in the introduction, the authors mention different kinds of regrets and other fundamental notions without describing the main ideas behind these concepts or how they relate in general. This may make some readers confused. Several times I also found myself wondering how different claims relate together, e.g., why are "certain Markov chains" important, how are they used in the proofs, and what their stationary distributions relate to. This makes it difficult to follow the authors' thought processes, and the reader is forced to rely on the explanations in the appendix, which contradicts the idea of the main text being self-contained.

The work is also highly technical, which is an inherent problem of the framework of sequential decision-making. The problem is further aggravated by the plethora of notation, including upper and lower indexing. I believe a table of symbols similar to Table 1 in Farina et al. 2021a would be convenient. Moreover, the work is absolutely void of examples, despite containing many definitions and advanced concepts. The authors may consider providing some to help the readers familiarize themselves with the introduced notions.

The main result the authors present is indeed interesting. Still, after reading through the appendix, my impression is that many of the constructions and proofs (with the important exception of Lemma B.7) are directly inspired by previous works, especially the papers by Farina et al. May the authors elaborate what were the main difficulties they encountered when extending the results into the framework of optimistic regret minimizers?

Farina et al. also define approximation of extensive-form correlated equilibrium using regret minimizers on the set of deterministic sequence-form strategies (Theorem 3.7 in Farina et al. 2021a), and moving to the set's convex hull (i.e., mixed strategies) will result in a guarantee with high probability only. According to Section 3.1, the sequence-specific regret minimizers are constructed from the regret minimizers for the mixed strategies (Q^i_j). Should not the approximation mentioned, e.g., already in the abstract, be hence also probabilistic?

I understand that using the technique of Chen and Peng would result in a non-polynomial degradation in stability. May I ask how this would affect the overall rate of convergence to extensive-form correlated equilibrium and computational complexity of the algorithm computing fixed points? More specifically, would the convergence rate contain game-specific exponential constants and otherwise be similar, or would it change completely?

Lastly, I wonder how fast one iteration of the OMWU-based dynamic is, compared to the alternatives (regret-matching+ and vanilla MWU)?

Nits:
The proof of proposition 3.3 is missing a reference (it reads "??").
Equation 45 is missing a comma.
At the beginning of Section 3, should not (2) be 2) instead?
The word "coarse" is missing in Corollary C.7.


**Summary Of The Paper:**

This paper presents an uncoupled no-regret learning dynamic provably converging to the extensive-form correlated equilibrium in general-sum n-player extensive-form games. The central claim of the paper is that the introduced approach results in the convergence rate of O(T^{-3/4}) to the equilibrium and hence supersedes the recent algorithm of Farina et al. which converges with rate O(T^{-1/2}). To achieve this, the authors follow the construction presented by Farina et al., but employ stable and predictive regret-minimizers from the class of Optimistic Follow the Regularized Leader algorithms instead of CFR/regret-matching used by Farina et al. The main portion of the paper is then dedicated to the theoretical analysis of the dynamic to arrive at the desired convergence rate. To this end, the authors first study how to construct stable-predictive regret minimizers for the convex hull of the set of trigger deviation functions for a given sequence, and consequently also for the composite regret minimizer. This enables them to bound the overall incurred regret. The construction also requires that every trigger deviation function admits a fixed point that is efficiently computable by a stable oracle. The stability is hence of the authors' interest in the second part. The paper is concluded by an experimental evaluation of the algorithm that shows that it is superior or performs on par with the algorithm of Farina et al. instantiated with regret matching or vanilla multiplicative weights updates.

**Summary Of The Review:**

Overall, I am slightly in favor of accepting this work. The problem is worth studying, the main result the authors present is exciting, and the empirical results support their claims. My main concerns relate to the presentation of this work. The text is extremely technical and notation-heavy, and I believe many concepts should be more adequately explained for a reader less familiar with no-regret learning to understand this work. The manuscript also completely lacks any examples. I remain unsure if a better exposition is possible given the ICLR's strict page limit and if this work would not be better suited for journals or conferences that allow longer narratives.

---

> ### Author Response · Authors · 2021-11-13
> **Response to Reviewer zmM5**
>
> We are grateful to the reviewer for the comments and the suggestions. Below we give our response.
>
> --- "May the authors elaborate what were the main difficulties they encountered when extending the results into the framework of optimistic regret minimizers?":
>
> We have created a separate subsection in the introduction to better highlight our own contributions. In particular, in the third paragraph of the Contributions section (which has been highlighted) we have included additional discussion on our technical contributions and how our approach differs from prior results.
>
> --- "Moreover, the work is absolutely void of examples, despite containing many definitions and advanced concepts.":
>
> We have included an example and an illustration of a simple EFG (highlighted in pp. 4) to help the reader familiarize with the basic concepts in EFGs.
>
> --- "I believe a table of symbols [...] would be convenient.":
>
> We recognize that the notation might be heavy, but there is an inherent complexity in the considered setting. We have included a table of notation (in pp.4 of the revised version) to ameliorate this concern.
>
> --- "Should not the approximation mentioned [...] be hence also probabilistic?":
>
> Our regret bounds are not probabilistic since we do not employ sampling. This is relatively common, especially in the literature on normal-form games. We remark that for our acceleration result the typical sampling approach appears to create non-trivial issues. In particular, using a standard concentration bound, as employed by Farina et al., leads to an error of $O(\sqrt{T})$ in the cumulative regret, which is prohibitive in our setting---the sampling error would essentially negate the acceleration. We are not aware of an approach that would handle this issue.
>
> --- "I understand that using the technique of Chen and Peng would result in a non-polynomial degradation in stability. May I ask how this would affect the overall rate of convergence to extensive-form correlated equilibrium and computational complexity of the algorithm computing fixed points?":
>
> First of all, we point out that Chen and Peng only studied normal-form games. We observed that an extension of their techniques based on the Markov chain tree theorem, without our refined techniques, would lead to an exponential dependence with respect to the depth of the tree. While this does not affect the complexity of computing the fixed point per iteration, the obtained bound for the rate of convergence would be exponential in the description of the game. We believe that this would substantially weaken the importance of the acceleration in terms of $T$, both in theory and in practice. But the reviewer is correct in noting that in terms of $T$ both methods lead to the same rate of convergence.
>
> --- "I wonder how fast one iteration of the OMWU-based dynamic is, compared to the alternatives":
>
> In Section 4 it is noted that one iteration of OMWU costs the same (up to constant factors) compared to the other methods. We have highlighted this point further, as we agree that it is an important consideration.

---

> > ### Comment · Reviewer_zmM5 · 2021-11-24
> > **Thank you for responding**
> >
> > Thank you for responding and clearing some of my misunderstandings, especially the problem of the probabilistic bound. I apologize for answering so late, but I had some personal matters I had to attend to first.
> >
> >
> > -- The technique of Chen and Peng
> >
> > If I understand your comment well, it means that to prove Theorem 1.1, you actually do not need lemma B.8, because here you speak of an asymptotic bound. To end up with such a bound, a simple extension of the Chen and Peng's technique would suffice. I agree, though, that the ability to bound the per-iteration regret such that all the constants depend on the game polynomially is a nice result. Could you mention how would the exponential constants look like? Also, I fail to see what do you mean by saying that the exponential constants would "weaken the importance of the acceleration ... in practice". I thought that running the OMWU does not depend on the techniques in lemma B.8 at all, i.e., it is a purely theoretical result. Or am I missing something?
> >
> >
> >  -- Computational costs
> >
> > Regarding the computational costs of OMWU, I understand that the complexity is the same, as I remember that you mentioned in the appendix that "the complexity of each iteration of the described dynamics is analogous to that in (Farina et al., 2021a)". This was not the point of my question though, I apologize for not being explicit. I was curious exactly how important the "constant factors" you mentioned are. How fast in terms of milliseconds is it to run one iteration of OMWU / MWU / RM+ on the experimental domains you used? Because my concern is that despite the EFCE gap decreases relatively faster per-iteration, as your experiments show, in case e.g. RM+ would be significantly faster, it would be still more time efficient to run RM+ instead of OMWU to reach a desirable approximation quality.

---

> > > ### Author Response · Authors · 2021-11-25
> > > **Response to Reviewer zmM5**
> > >
> > > We really thank the reviewer for the additional feedback. Below we address the questions raised.
> > >
> > > --- "If I understand your comment well, it means that to prove Theorem 1.1, you actually do not need lemma B.8, because here you speak
> > > of an asymptotic bound."
> > >
> > > Lemma B.8 is crucial for the proof of Theorem 1.1, since the statement of Theorem 1.1 asserts a rate of convergence that depends *polynomially* on the size of the game.
> > >
> > > --- "Could you mention how would the exponential constants look like?"
> > >
> > > Extending the techniques of Chen and Peng gives a dependence which in the worst case is proportional to the *product* of the number of actions at each of the decision points of the game. Note that this bound is *exponential* in the game tree size. Instead, our dependence scales with the *sum* of the number of nodes in the game tree, which is linear in the representation of the input game tree. We will clarify this point in the final version.
> > >
> > > --- "Also, I fail to see what do you mean by saying that the exponential constants [...] Or am I missing something?"
> > >
> > > The reviewer is correct in that the per-iteration running time of OMWU is not affected by Lemma B.8. However, the overall number of iterations required to provably reach an approximate equilibrium crucially depends on whether the dependence on the game size is polynomial or exponential (which is guaranteed by Lemma B.8). In particular, with an exponential dependence on the game size, we wouldn’t be able to guarantee—a priori—that only a polynomial number of iterations (in the game size) are necessary to reach a, for instance, 0.01-approximate EFCE.
> > >
> > > --- "I was curious exactly how important the "constant factors" you mentioned are. How fast in terms of milliseconds is it to run one iteration of OMWU / MWU / RM+ on the experimental domains you used?"
> > >
> > > The differences between the running times of the three algorithms are negligible (we will mention this in the final version). We computed the average running time of one iteration of the three algorithms in each of the three benchmark games (across 10k iterations for Kuhn poker and Sheriff, and 1k iterations for Liar’s dice, the same number of iterations as the plots in the manuscript):
> > >
> > > Kuhn poker (MWU): 0.7274 milliseconds/iteration\
> > > Kuhn poker (OMWU): 0.7319 milliseconds/iteration\
> > > Kuhn poker (RM+): 0.7416 milliseconds/iteration
> > >
> > > Sheriff (MWU): 3.1715 milliseconds/iteration\
> > > Sheriff (OMWU): 3.2264 milliseconds/iteration\
> > > Sheriff (RM+): 3.1845 milliseconds/iteration
> > >
> > > Liar’s dice (MWU): 634.4065 milliseconds/iteration\
> > > Liar’s dice (OMWU): 634.3065  milliseconds/iteration\
> > > Liar’s dice (RM+): 632.3201 milliseconds/iteration
> > >
> > > These results are consistent with the fact that MWU, OMWU and RM+ boil down to a very similar sequence of operations at each iteration (with the difference that MWU/OMWU exponentiate the cumulated regrets before normalizing, while RM+ thresholds with zero the cumulated regrets instead before normalizing).

---

### Official Review · Reviewer_NoAR · 2021-11-03

**Correctness:** 3
**Technical Novelty And Significance:** 3
**Empirical Novelty And Significance:** 2
**Recommendation:** 6
**Confidence:** 3

**Main Review:**

## Strengths
The problem that the authors considered is an important problem for no-regret learning dynamics in games. Compared to existing literature on accelerated learning dynamics for correlated equilibria (CE) or coarse correlated equilibria in normal-form games, the convergence rate for extensive-form game is relatively less understood.

This paper shows an improved convergence rate to extensive-form correlated equilibrium (EFCE): when the game is played for T repetitions, the correlated distribution of play for all players is an $O(T^{-3/4})$-approximated EFCE, which improves upon the previous best rate $O(T^{-1/2})$. To me this is a good and important theoretical contribution to the existing literature.

A main conceptual contribution is to establish a connection between the optimistic regret minimization to $\phi$-regret minimization problem. A main technical contribution is a characterization of the stability of certain fixed point strategies through a refined
perturbation analysis of the structured Markov chain. Other than that the proof utilized previous results in Farina et al.
(2021a), which casts the convergence to an EFCE as a $\phi$-regret minimization problem, and existing framework on optimistic regret minimization.

Numerical simulations are provided in support of the theoretical results.

## Weakness

First I think the summary of related works on no-regret learning dynamics for normal-form games are missing some most recent results. For example in [Daskalakis, Fishelson, Golowich, Near-Optimal No-Regret Learning in General Games, 2021] it is shown that one can achieve a O(1/T) convergence rate to coarse correlated equilibrium in multi-player general-sum games. Secondly I don't see much discussion on the technical difficulty of why the paper's framework could not be applied to normal-game. Given the many other existing rates for normal-games, it would be good to add more details on this point.

Regarding the technical contributions, the main concern is about novelty given that the main framework is built on previous results in Farina et al. (2021a) which casts the convergence to an EFCE as a $\phi$-regret minimization problem, and existing framework on optimistic regret minimization as in Gordon et al. (2008).




**Summary Of The Paper:**

This paper proves a faster no-regret learning dynamics for extensive-form correlated equilibrium (EFCE) in multiplayer general-sum imperfect-information extensive-form games. When the game is played for T repetitions according to the accelerated dynamics, the correlated distribution of play for all players is an $O(T^{-3/4})$-approximated EFCE. This improves upon the previous best rate which is $O(T^{-1/2})$ for extensive-form games.

**Summary Of The Review:**

Overall I think this paper provides nice theoretical contribution with an improved rate for no-regret learning dynamics that converge to extensive-form correlated equilibrium. The paper is well-written. However currently the related work is missing some recent relevant results, and on the technical contribution side there should be some more details differentiating the new results and existing frameworks.

Post rebuttal update: I have read the authors' response and will keep the original score.

---

> ### Author Response · Authors · 2021-11-13
> **Response to Reviewer NoAR**
>
> We are grateful to the reviewer for the comments and the suggestions. Below we give our response.
>
> --- "First I think the summary of related works on no-regret learning dynamics for normal-form games are missing some most recent results.":
>
> We have included a discussion about the recent result of Daskalakis, Fishelson, and Golowich (Near-Optimal No-Regret Learning in General Games). While this shows a rate of convergence of $\widetilde{O}(1/T)$, it holds for coarse correlated equilibria and in normal-form games. There are several additional challenges to overcome in our setting. First, their analysis crucially relies on the fact that the optimization is performed on the simplex, using the closed-form softmax-type structure of OMWU. Thus, it is not known whether their result can be extended even for normal-form coarse correlated equilibria in extensive-form games. Moreover, the learning dynamics for correlated equilibria, even in normal-form games, are substantially more involved as they include a fixed point operation. This is part of the reason why the result of Syrgkanis et al. was only recently extended for correlated equilibria (in normal-form games). We also point out to the reviewer that at the time of our submission this work was not published.
>
> --- "On the technical contribution side there should be some more details differentiating the new results and existing frameworks.":
>
> We have created a separate subsection in the introduction to better highlight our own contributions. In particular, in the third paragraph of the Contributions section (which has been highlighted) we have included additional discussion on our technical contributions and how our approach differs from prior results.

---

### Official Review · Reviewer_4e5F · 2021-11-05

**Correctness:** 4
**Technical Novelty And Significance:** 4
**Empirical Novelty And Significance:** Not applicable
**Recommendation:** 8
**Confidence:** 2

**Main Review:**

The paper is well written (see specific comments and the concern below). The proofs seem correct (I did not check the appendices in detail). Providing an accelerated convergence rate for multiplayer extensive form games (and in particular proving the counterpart of what is known for normal-form game) is a valuable contribution. My main concern is that in the current form the paper is difficult to read. In particular, the proposed algorithm should be at least described in the main text (see main comments). Furthermore, some parts should be better introduced, e.g. the \Phi-regret minimization framework (see specific comments).

Main comments:

-It is hard to find what is the final algorithm mentioned in Theorem 1.1 by reading the main text and even by looking at Appendix B. Thus without the big picture Section 3 looks a bit like a collection of unrelated technical results.

-Could you state clearly the complexity of the proposed algorithm (it seems that it scales at least quadratically with the size of the game?) and polynomial dependence on the size of the game hidden in Theorem 1.1. This is important since usually the size of the game is very large.

Specific comments:
-P1 Th 1.1: Is there any lower bound on the rate? For example, is there hope to improve the rate to O(1/T) as for two players? You should also clearly refer to the description of the algorithm in the paper.

-P3 End of section 1: Why normal-form coarse correlated equilibria is solution concept much less appealing than EFCE?

-P4, below (1): typo \cJ^{(i)}

-P5, Theorem 2.5: \psi^(i)-regret is not defined at this point.

-P6, Definition 3.1: does the point x_t necessarily coincide with the point played by the regret minimizer algorithm? Ok maybe you should describe the framework of Gordon et al. (x_t is the fixed point of a \phi_t obtained by another regret minimizer on an auxiliary task?).

-P6, Theorem 3.2: Maybe you should define clearly what you mean by a regret minimizer R_\Phi
for the set of transformation \Phi.

-P6, end of Section 2: could you detail what you mean by entropic regularizer on \cX?

-P9, end of Section 3: Could point out precisely where is the final algorithm in Appendix B.

-P9, Section 4: At this point, it is not clear at all what the OMWU algorithm is. Did you also experiment on Leduc Poker? Because it is usually the setting where there is a gap between multiplicative weights type of algorithms and RM+ (ith alternating updates).

-P13, top of the page: \cJ is in the introduction the set of information set. Using the same notations for the sequential decision making setting is a bit confusing. Could you

-P13, (13): is an observation node always followed by an action node?

-P16, above (24): typo

-P18, above (32): norm 1

**Summary Of The Paper:**

The authors consider the problem of finding an approximate extensive-form correlated equilibrium (EFCE) of a finite general-sum multiplayer extensive-form game under the perfect recall assumption. They propose an accelerated version via optimism of the algorithm by Farina et al. (2019a). Precisely it combines the framework by Gordon et al. (2008) for \Phi-regret minimization with the framework of optimistic regret minimization adapted to extensive form game. They prove that when all agents play T repetitions of the game according to the proposed algorithm the correlated distribution of play is an O(T ^−3/4 )-approximate EFCE where O hides quantities polynomial in the size of the game. The main technical point to obtain this result is to characterize the stability of certain fixed point strategies through a refined perturbation analysis of a structured Markov chain. They also provide preliminary experiments on simple two or three-players games.

**Summary Of The Review:**

See Main Review

---

> ### Author Response · Authors · 2021-11-13
> **Response to Reviewer 4e5F**
>
> We are grateful to the reviewer for the comments and the suggestions. Below we give our response.
>
> --- "It is hard to find what is the final algorithm mentioned in Theorem 1.1":
>
> We have included a figure (Figure 2) illustrating the central components of the algorithm, located in the beginning of Section 3.1. These components are further explained in Appendix B.
>
> --- "Could you state clearly the complexity of the proposed algorithm":
>
> To avoid an overly complicated expression, we did not explicitly give the polynomial dependence of the rate of convergence on the parameters of the game in Theorem 1.1, as is common in the line of work on EFGs. We note that an exact bound follows directly from our intermediate results.
>
> --- "Why normal-form coarse correlated equilibria is [...] much less appealing than EFCE?":
>
> We have included a reference to the work of Gordon, Greenwald, and Marks (No-regret learning in convex games), where there is a detailed discussion on why normal-form coarse correlated equilibrium constitutes a weak and rather unrealistic notion of hindsight rationality, especially in a sequential decision making setting. To quote their work, "a no-external-regret learner can consistently observe that its average pay-off per trial would have been higher if it had chosen action $a'$ every time that it actually played $a$, and yet never switch to playing action $a'$ in these situations". Recall that no-external-regret dynamics are associated with convergence to coarse correlated equilibria.
>
> --- "Theorem 2.5: $\Psi^{(i)}$-regret is not defined at this point":
>
> We have rearranged the preliminaries so that $\Psi^{(i)}$-regret is defined before the scope of Theorem 2.5.
>
> --- "Definition 3.1: does the point $\mathbf{x}^t$ necessarily coincide with the point played by the regret minimizer algorithm?"
>
> In Definition 3.1 $\mathbf{x}^{t}$ is a fixed point of the transformation $\phi^t$, where $\phi^t$ is the output of the regret minimizer operating over the set of transformations $\Phi$. Note that the way we compute the fixed point does \emph{not} affect the regret bounds (although it does affect the complexity per iteration).
>
> --- "Theorem 3.2: Maybe you should define clearly what you mean by a regret minimizer $\mathcal{R}_\Phi$ for the set of transformation $\Phi$."
>
> Assuming that a set $\Phi$ is convex and compact, an external regret minimizer for the set $\Phi$ can be understood under the framework described in our preliminaries, applicable for an arbitrary convex and compact set $\mathcal{X}$.
>
> --- "Could you detail what you mean by entropic regularizer":
>
> We have formally defined the entropic regularizer on the simplex.
>
> --- "At this point, it is not clear at all what the OMWU algorithm is. Did you also experiment on Leduc Poker":
>
> In our experimental section, by performing Optimistic Multiplicative Weights Update (OMWU) we mean that every ``local'' regret minimizer is instantiated using OMWU. The same convention applies for MWU and RM$^+$. We did not compare the algorithms in Leduc poker, but we will try to conduct additional experiments on Leduc.
>
> --- "$\mathcal{J}$ is in the introduction the set of information sets. Using the same notations for the sequential decision making setting is a bit confusing":
>
> We see how using the same symbol $\mathcal{J}$ in the setting of sequential decision making can potentially cause some confusion, but note that there is a one-to-one correspondence between the two settings. This notation is also chosen in order to be consistent with prior works.
>
> --- "Is an observation node always followed by an action node?":
>
> We can always assume without loss of generality that an observation node for a given player is indeed always followed by an action (decision) node. (If this is not the case, one can always equivalently "merge" multiple observation nodes into a single observation node.)
>
> --- "Is there any lower bound on the rate? For example, is there hope to improve the rate to $O(1/T)$ as for two players?"
>
> There is a lower bound of $\Omega(1/T)$ within the no-regret framework. A very recent result by Daskalakis, Fishelson, and Golowich (Near-Optimal No-Regret Learning in General Games) establishes a rate of $\widetilde{O}(1/T)$, but this applies for coarse correlated equilibria and in normal-form games. While this gives some hope that our results could be further improved in the future, there are substantial challenges to be overcome before one can use their techniques in our setting. We have included a discussion about this point in the related work section.

---

### Decision · Program_Chairs · 2022-01-20

**Decision:**

Reject

**Comment:**

This paper builds upon existing works to prove that learning (correlated) equilibrium can be fast, i.e., faster than \sqrt{n} even in extensive form games.

Three reviewers are rather lukewarm, and one reviewer is more positive (but seems less confident in his score). The two major criticisms is that this paper is very difficult to read and that the results might seem rather incremental with respect to the literature.

I tend to agree with both points but the paper still as merits: the reason is that extensive form games are intrinsically way harder than normal form games and they more or less all have a burden of notations. We agreed  that the authors actually did some efforts to make it fit within the page limit. but another a conference or a journal would have been better suited than ICLR.

Our final conclusion is that the result is interesting yet maybe not breathtaking for the ICLR community; we are fairly certain that another venue for this paper will be more appropriate and that it will be accepted in the near future (I can only suggest journals based on the large amount of content and notations, such as OR, MOR, or GEB - yet, conferences such as EC should be more scoped too) . It does not, unfortunately, reach the ICLR bar.